# PCGS: Deblurring 3D Gaussian Splatting with Patch Comparison

**Yilong Li** [1]  **Bo Pang** [1]  **Zhongtao Wang** [1]  **Mai Su** [1]  **Yisong Chen** [1]  **Chengwei Pan** [2]  **Meng Gai** [1]  **Fei Zhu** [1]
**Guoping Wang** [1]

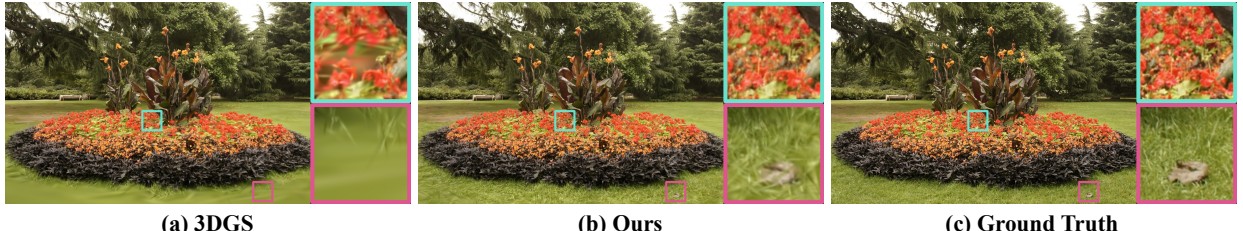

| (a) 3DGS | (b) Ours | (c) Ground Truth |
|---|---|---|

*Figure 1.* Densification is a crucial component of 3D Gaussian Splatting (3DGS). Constraints imposed by view-space positional gradients, while aiding adaptive control of the Gaussians, can also unintentionally lead to artifacts, such as those seen in **(a)** (e.g., the grass and the flowers). Our approach **(b)** mitigates this issue by introducing a patch comparison strategy and growth control strategy, which guide the Gaussians toward effective densification in blurry regions. For reference, **(c)** shows the ground truth, providing a clear benchmark for evaluating the quality of the results.

## Abstract

Recent neural methods, such as 3D Gaussian Splatting, have achieved state-of-the-art rendering quality and speed. However, these methods frequently encounter challenges in regions with overlapping Gaussians, leading to blurring and artifacts in the rendered images. We observed that widely used view-space positional gradients are insufficient for handling such circumstances. To address this, we introduce PCGS, a Patch Comparison Gaussian Splatting method to control the densification of corresponding Gaussians adaptively. Specifically, PCGS divides the rendered image into patches and identifies those with significant errors by comparing the loss between the rendered and ground truth images. Additional densification operations are then applied to the Gaussians in these error-prone regions. Furthermore, to prevent over-densification and redundant Gaussians, we design a Gaussian control strategy to regulate the densification process. Specifically, we set a Gaussian number budget that dynamically changes according to the progress of densification, and sample the Gaussians required for each densification step based on their importance scores. Our method results in significantly fewer artifacts and less blur while maintaining a Gaussian count approximately equal to that of 3DGS. Extensive experiments on multiple standard benchmarks demonstrate the superiority of our approach.

## 1. Introduction

Photorealistic novel viewpoint synthesis has long been a central challenge in computer graphics and vision. The goal is to generate new images of a scene from arbitrary viewpoints using multiple input views. The advent of Neural Radiance Fields (NeRF) (Mildenhall et al., 2021) marked a significant breakthrough in novel viewpoint synthesis by utilizing MLPs (multi-layer perceptrons) to implicitly record the light field. Despite its success, typical NeRF-based methods require a large number of MLP queries, resulting in slow training and rendering.

Recently, 3D Gaussian Splatting (3DGS) (Kerbl et al., 2023) has gained significant attention due to its high rendering quality and fast training speed. By employing a set of 3D Gaussian ellipsoids along with an adaptive densification scheme, 3DGS can effectively produce high-quality renderings. However, current Gaussian Splatting models may still produce blur and artifacts for challenging scenes, as demonstrated in Figure 1.

Our key observation is that the traditional Gaussian Splatting

[1]National Key Laboratory of Intelligent Parallel Technology, Peking University, Beijing, China [2]School of Artificial Intelligence, Beihang University, Beijing, China. Correspondence to: Yisong Chen <chenyisong@pku.edu.cn>, Fei Zhu <feizhu@pku.edu.cn>.

*Proceedings of the 43rd International Conference on Machine Learning*, Seoul, South Korea. PMLR 306, 2026. Copyright 2026 by the author(s).

training process introduces a substantial imbalance in the contributions of individual Gaussians, as illustrated in Figure 2. Specifically, a small subset of Gaussians often exerts a dominant influence over a large portion of pixels, while other pixels are shaped by contributions from far more Gaussians. In regions where Gaussians are sparsely distributed, the rendered image may appear blurry due to insufficient information to accurately capture the scene. Conventional pixel-wise loss is often insensitive to these blurry regions, leading to small view-space positional gradients for the corresponding Gaussians. As a result, the densification process is hindered, making it challenging for Gaussians to clone or split effectively. Conversely, the distribution of Gaussian in some areas is redundant, resulting in unnecessary computational and storage costs. Accordingly, a natural and intuitive question arises: How can we effectively balance the density of Gaussians in the scene to hit a sweet spot of the rendering quality and efficiency?

We propose Patch Comparison Gaussian Splatting (PCGS), a novel 3D Gaussian framework that effectively alleviates the blurring issues through patch comparison. Particularly, PCGS first uses the difference between the rendered and ground truth images to identify error-prone pixels. It then partitions the rendered image into patches. The errors of the member pixels are aggregated onto the corresponding patches to help identify whether a Gaussian needs densification. Traditional methods focus on pixel-based loss functions, which are often insensitive to patch-level blurring and artifacts in the rendered images. In contrast, our patch comparison strategy exploits spatial coherence: blur artifacts typically span contiguous regions rather than isolated pixels. By aggregating pixel-level errors within patches, we achieve more robust identification of under-reconstructed regions while reducing sensitivity to noise. Furthermore, the densification strategy used in the original 3DGS often leads to redundant Gaussians. To address this issue, we introduce a Gaussian growth control mechanism that limits the number of newly added Gaussians at each densification step and selects candidates for densification based on their importance scores. In summary, the contributions of our PCGS are as follows:

- We introduce a patch comparison strategy to address the blur and artifacts issue. The patch comparison effectively identifies poorly reconstructed regions and directs the densification process accordingly.

- We propose a Gaussian growth control strategy that selectively determines which Gaussians participate in each densification step, thereby improving rendering quality while reducing redundancy.

- Extensive experiments demonstrate the superiority of our method both quantitatively and qualitatively.

## 2. Related Work

### 2.1. Traditional View Synthesis

Earlier methods (Szeliski, 1997; Shum & He, 1999) generated novel views by interpolating known neighboring views. However, these approaches required extensive captures and had limited ability to synthesize new views. Later techniques (Chaurasia et al., 2013; Penner & Zhang, 2017) leveraged 3D structural information extracted from input images. Nonetheless, the presence of complex structures in the scene can hinder the accurate recovery of corresponding 3D information.

### 2.2. Neural Radiance Fields

With the advancement of deep learning, neural networks have been increasingly applied to novel view synthesis. The primary approach involves using neural networks as implicit functions to represent scene structures. Common neural networks used in this domain include occupancy networks(Mescheder et al., 2019), and signed distance functions (SDF)(Park et al., 2019).

Neural radiance fields (NeRF) (Mildenhall et al., 2021) have achieved photorealistic rendering by constructing radiance fields with volume rendering. A series of NeRF-based works have since emerged to address various challenges, including improving training speed (Müller et al., 2022; Gupta et al., 2023), enabling real-time rendering (Hedman et al., 2021; Garbin et al., 2021), enhancing rendering quality (Barron et al., 2022; Park et al., 2023), improving generalization (Yu et al., 2021; Chibane et al., 2021), and handling dynamic scenes (Pons-Moll et al., 2021; Fridovich-Keil et al., 2023). Among these methods, InstantNGP (Müller et al., 2022) introduces multi-resolution hash tables to enable fast training and real-time rendering. MipNeRF360 (Barron et al., 2022) effectively compresses unbounded scenes into a distorted space, allowing for efficient sampling by mapping distant points into a limited space.

### 2.3. 3D Gaussian Splatting

More recently, 3D Gaussian Splatting (3DGS) has emerged as a state-of-the-art method. It uses anisotropic Gaussians to represent scenes and employs a differentiable tile-based rasterizer. This approach eliminates the need to query MLPs in a ray-tracing fashion, significantly accelerating the rendering of novel views. This method has been rapidly generalized to several domains, including editing (Chen et al., 2024; Wang et al., 2024; Zhou et al., 2024), language embedding (Shi et al., 2024; Qin et al., 2024), mesh extraction (Guédon & Lepetit, 2024; Huang et al., 2024), SLAM (Yan et al., 2024; Matsuki et al., 2024), and dynamic scenes (Luiten et al., 2024; Yang et al., 2024; Wu et al., 2024; Yang et al., 2023; Duan et al., 2024).

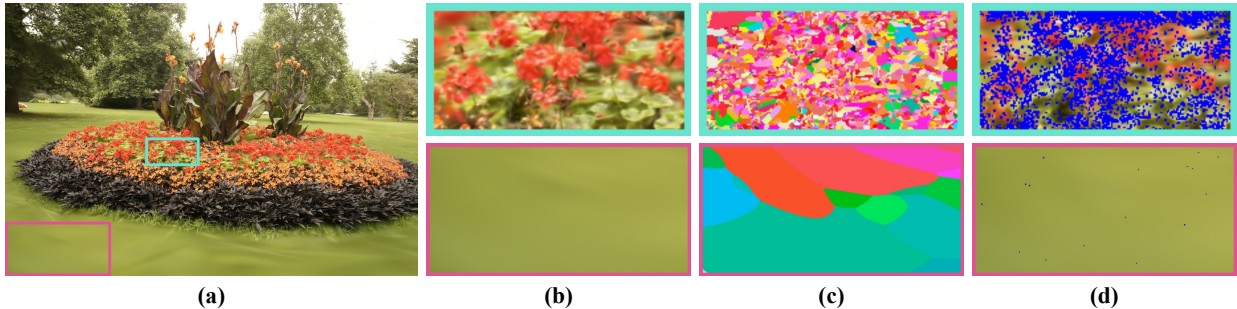

**(a)**        **(b)**        **(c)**        **(d)**

*Figure 2.* For images rendered with the original 3DGS that exhibit artifacts (a), we select two regions with different reconstruction qualities (b). We then visualize the most contributing Gaussians per pixel (c) and the 2D projection of all visible Gaussians (d). Specifically, we assign a unique color to each Gaussian based on its hash value, making connected regions in (c) represent the influence range of the same Gaussian. Blue dots in (d) indicate the projection centers of Gaussians. This reveals that **artifact-prone areas are often dominated by a few large Gaussians, while detailed regions contain an excess of Gaussians.**

Despite its advantages, 3DGS is prone to blur issues. This can be partially attributed to the ineffective splitting of large Gaussians, as their view-space position gradients fall below the threshold. To address this, previous work has attempted to increase their view-space position gradients(Ye et al., 2024; Zhang et al., 2024b;a) or replace the view-space position gradient with an alternative metric(Rota Bulò et al., 2024; Zhou & Ni, 2025). However, directly increasing the view-space position gradient may cause some Gaussians with previously sub-threshold gradients to exceed the threshold, resulting in forced densification. Some methods (Fang & Wang, 2024; Niemeyer et al., 2025) attribute artifacts to the lack of initial point clouds in affected regions, which hinders effective densification. To address this, they reinitialize the point cloud to obtain a better distribution. However, this approach requires the pre-training of Gaussians or even NeRF for knowledge distillation, which increases the overall training time. Steep-GS(Wang et al., 2025) designs specific functions to determine whether a Gaussian should be split and to predict the positions of the resulting sub-Gaussians. Scaffold-GS(Lu et al., 2024) employs an anchor-based decoding scheme to control Gaussian attributes at different viewpoints. Both methods modify the original 3DGS framework but fail to address the issue of Gaussian-induced artifacts.

Compared to other methods that inadvertently affect the cloning and splitting of Gaussians besides the large ones, our approach, PCGS, introduces a patch comparison mechanism to specifically identify and split large Gaussians in artifact regions without altering other Gaussians' position gradients. Additionally, we propose a growth control strategy to effectively regulate the Gaussian densification process.

## 3. Method

PCGS is designed to address blur issues and achieve high-quality rendering. An overview of our method is provided in Figure 3. In this section, we detail its components and the corresponding optimization process. First, we briefly review 3D Gaussian Splatting (Kerbl et al., 2023) in Section 3.1. Second, we introduce our patch comparison strategy, which effectively identifies regions that require densification in Section 3.2. In Section 3.3, we present our growth control strategy to regulate the densification of Gaussians to improve rendering quality.

### 3.1. Preliminary

3D Gaussian Splatting represents the scene using a set of anisotropic 3D Gaussian ellipsoids. Each Gaussian is parameterized by a covariance matrix $\Sigma$ and a 3D center position $\mu$:

$$G(x) = e^{-\frac{1}{2}(x-\mu)^T \Sigma^{-1}(x-\mu)} \tag{1}$$

where $\mu \in R^3$ and $\Sigma \in R^{3\times3}$. To ensure the covariance matrix remains valid and positive semi-definite during the optimization process, $\Sigma$ is decomposed into a scaling matrix $S$ and a rotation matrix $R$:

$$\Sigma = RSS^T R^T \tag{2}$$

To render the image, the 3D Gaussian function is projected to 2D using the view transformation matrix $W$ and the Jacobian $J$ of the affine approximation of the projective transformation:

$$\Sigma' = JW\Sigma W^T J^T \tag{3}$$

Subsequently, 3D Gaussian Splatting employs volumetric alpha blending to integrate the weighted appearances in depth order:

$$C = \sum_{i=1}^{N} c_i \alpha_i \prod_{j=1}^{i-1}(1-\alpha_j) \tag{4}$$

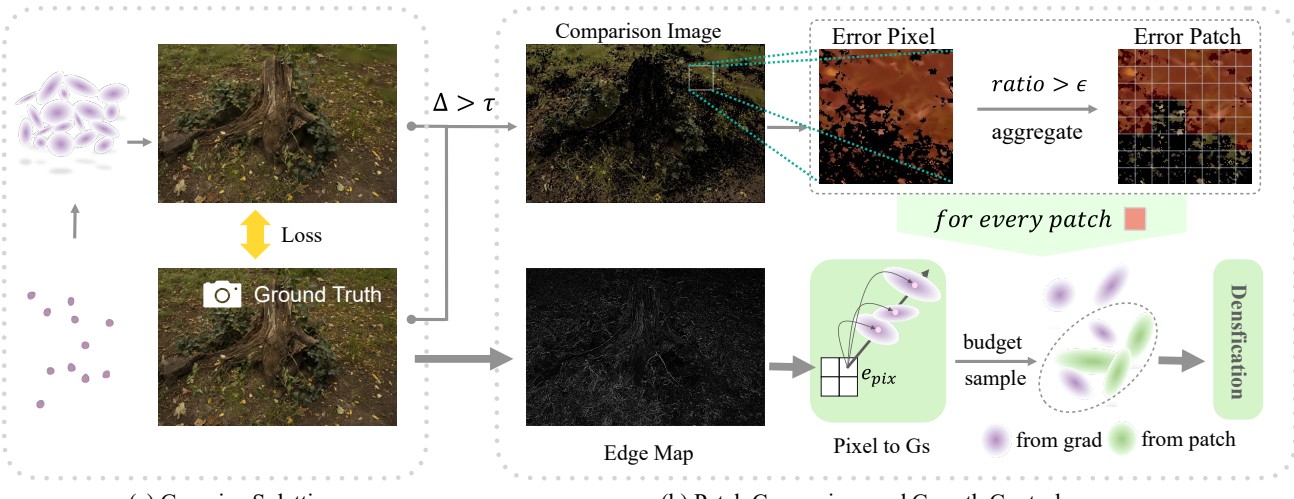

(a) Gaussian Splatting      (b) Patch Comparison and Growth Control

*Figure 3.* **Overview of PCGS.** (a) After initialization, we render 3D Gaussian points to an image and compare it with the ground truth. (b) We calculate the pixel-wise error between our image and the ground truth image. The pixels whose errors exceed a threshold $\tau$ are identified as "error-pixels". We then construct patches, and mark patches in which the ratio of "error-pixels" exceeds the threshold $\epsilon$ as "error-patches" (marked in red in the image). For each such patch, the Gaussians that predominantly influence the patch (marked in green) are combined with those selected based on the view-space position gradients (marked in purple) to form the candidate set. We then introduce a Gaussian growth control strategy that selects appropriate Gaussians (within the dashed circle) for densification based on their importance scores, which helps maintain a reasonable number of Gaussians while improving rendering quality.

where $c_i$ is the color of each point. and $\alpha_i$ is estimated by evaluating a 2D Gaussian with the covariance $\Sigma_{2D}$ multiplied by a learned per-point opacity $o_i$ The properties of the 3D Gaussian primitives are optimized using a photometric loss function between the rendered and ground truth images.

### 3.2. Patch Comparison

In this section, we first introduce the densification mechanism of 3DGS. During training, to better fit the scene, 3DGS accumulates the view-space positional gradients of each Gaussian between densification intervals. At each densification step, Gaussians with gradients exceeding the threshold are selected, and their maximum scaling vector value is checked to determine whether they should be cloned or split. Consequently, the primary cause of artifacts can be attributed to the Gaussians responsible for them failing to meet the densification requirements. An intuitive yet naive strategy to address this issue is to lower the densification threshold. We tested various thresholds, ranging from 2e-4 to 5e-5. Our observations show that while lowering the densification threshold offers marginal metric improvement, it leads to a significant increase in the number of Gaussians, as shown in Table 1. Notably, when the threshold is reduced to 5e-5, the generated Gaussians lead to out-of-memory errors. This indicates that naively lowering the densification threshold is not appropriate, and a wiser solution is desired.

To address the above issues, we propose a patch comparison strategy to identify the blurry regions and guide the Gaus-

*Table 1.* **The results of different thresholds on the MipNeRF360 Dataset.** While a reduced densification threshold yields marginal quality gains, it incurs a substantial rise in the number of Gaussians, causing out-of-memory errors as the threshold approaches 5e-5.

| threshold | PSNR ↑ | SSIM ↑ | LPIPS ↑ | Gs(M) |
|---|---|---|---|---|
| 2e-4 | 27.21 | 0.815 | 0.214 | 3.18 |
| 1.5e-4 | 27.33 | 0.820 | 0.202 | 4.93 |
| 1e-4 | 27.34 | 0.821 | 0.195 | 8.94 |
| 5e-5 | Out of Memory | | | |

sians to densify. First, we compute a pixel-level loss map representing the discrepancy between the rendered image and the ground truth. To objectively separate significant artifacts from minor rendering noise, we employ Otsu's method (Otsu et al., 1975) to automatically determine an optimal pixel-level threshold $\tau$. By maximizing the inter-class variance of the loss distribution, this adaptive threshold effectively labels pixels with prominent errors as error pixels. Next, to identify regions requiring densification (e.g., blurry areas or missing details), we partition the rendered image into multiple patches. For each patch, we calculate the error pixel ratio (i.e., the proportion of error pixels within the patch). Instead of using a fixed sensitivity threshold, we apply Otsu's method a second time to the distribution of these ratios across all patches. This allows us to adaptively identify a threshold $\epsilon$ that segregates error patches from well-rendered regions. Our subsequent densification step focuses exclusively on these identified error patches, ensuring

an efficient and scene-dependent refinement process.

However, relying solely on the spatial distribution of error patches is insufficient to guide Gaussian densification, as the relationship between patches and Gaussians is non-bijective. A single patch is typically influenced by a collective of Gaussians; consequently, a naive densification of all contributing primitives would lead to an unbounded explosion in Gaussian count, compromising rendering efficiency. Furthermore, selecting the Top-K contributors for densification introduces a heuristic dependency, as the optimal $K$ is highly sensitive to scene complexity and evolves unpredictably across training stages. Crucially, determining a universal $K$ for each pixel is non-trivial, as defining a threshold for "nearly equal" contributions remains an open challenge. Indiscriminate Top-$K$ densification could also disrupt the underlying scene geometry and increase CUDA implementation complexity. To eliminate this sensitivity, we propose a more targeted refinement strategy: for each pixel within an identified error patch, we pinpoint and densify only the Gaussian that yields the maximum alpha-weighted contribution:

$$\mathcal{G}^{select} = arg \max_i \omega_i,$$
$$where \ \omega_i = \alpha_i \prod_{j=1}^{i-1}(1 - \alpha_j) \quad (5)$$

Effectively, this strategy decomposes the conventional Top-K densification task into a sequence of single-Gaussian refinements, enabling fine-grained, scene-adaptive densification while eliminating the need for manual parameter tuning. To ensure that all major Gaussians contributing to artifacts are comprehensively addressed, we implement a consistency-preserving refinement strategy inspired by (Zhou & Ni, 2025). By constraining the newly cloned Gaussians to maintain an aggregate contribution equivalent to the original primitive—specifically through proportional opacity reduction—we facilitate a dynamic redistribution of influence weights within the patch. Consequently, for regions where multiple Gaussians contribute to a blur, our strategy operates through a principled iterative process: once the most dominant Gaussians are densified and their individual weights diminish, they naturally "unmask" other significant contributors within the Top-K set for subsequent targeting. This mechanism prevents selection saturation and allows the system to progressively identify and refine influential Gaussians in a sequential, self-evolving manner. For empirical validation of this "unmasking" mechanism, please refer to Appendix B.

### 3.3. Growth Control Strategy

The densification process in the original 3DGS introduces redundant Gaussians, as shown in Figure 2. The additional densification brought by our patch-based comparison strategy further aggravates this problem. A simple solution is to impose a hard upper bound on the total number of Gaussians. However, such a constraint causes the limit to be reached early in training due to the extra densification, leaving subsequent iterations to merely fine-tune Gaussian attributes. As a result, geometry learning becomes incomplete, and the overall training is unnecessarily prolonged.

To enable smoother growth, we introduce a per-step Gaussian budget $A(t)$ that gradually increases the number of Gaussians:

$$A(t) = S + (B - S) * \frac{log(1 + k * t)}{\log(1 + k)},$$
$$where \ t = \frac{I - I_s}{I_f - I_s} \quad (6)$$

Here, $B$ denotes the final Gaussian budget, $S$ represents the number of points reconstructed by SfM, and $k$ controls the degree of exponential interpolation (we set $k = 9$). $I$, $I_s$, and $I_f$ denote the current, starting, and final iterations of the densification process, respectively. The logarithmic schedule encourages more Gaussian cloning and splitting during the early stages to better capture scene geometry, while naturally slowing the growth in later stages to prevent overfitting once sufficient Gaussians have been established.

Once the Gaussian budget for each densification step is established, the subsequent challenge is to identify the most critical candidate Gaussians for refinement. Simple uniform random sampling is suboptimal, as rendering artifacts exhibit significant spatial heterogeneity, with varying degrees of perceptual impact across different regions. Recognizing that the human visual system (HVS) is inherently sensitive to structural boundaries and high-frequency details, we leverage edge maps to guide the selection process. Specifically, we prioritize Gaussians located in perceptually salient areas by back-projecting the edge information into the 3D Gaussian space:

$$e_i = \sum_{p \in P} e_{pix} \quad (7)$$

where $e_i$ denotes the edge value estimated from the edge map for Gaussian $i$, $P$ represents the set of pixels influenced by this Gaussian, $e_{pix}$ denotes the edge value of pixel $p$. By accumulating the edge values of all pixels influenced by each Gaussian, we obtain the desired **importance score**.

We follow the original 3D Gaussian Splatting and use a combination of L1 and SSIM loss as our loss function.

$$\mathcal{L} = (1 - \lambda_1)\mathcal{L}_1 + \lambda_1 \mathcal{L}_{SSIM} \quad (8)$$

## 4. Experiments

We conducted both qualitative and quantitative evaluations on three real-world scene datasets to demonstrate the effec-

*Table 2.* **Quantitative comparisons on the Mip-NeRF360, Tanks&Temples and Deep Blending Datasets.** All methods were trained on the same data. We report the PSNR, SSIM, and LPIPS metrics. INGP-Base refers to the basic configuration(Kerbl et al., 2023), while INGP-Big represents a slightly larger network(Kerbl et al., 2023). The best , second best , and third best results are highlighted in red, orange, and yellow, respectively.

| Dataset | Mip-NeRF360 | | | Tanks&Temples | | | Deep Blending | | |
|---|---|---|---|---|---|---|---|---|---|
| Method | PSNR↑ | SSIM↑ | LPIPS↓ | PSNR↑ | SSIM↑ | LPIPS↓ | PSNR↑ | SSIM↑ | LPIPS↓ |
| Plenoxels(Fridovich-Keil et al., 2022) | 23.08 | 0.626 | 0.463 | 21.08 | 0.719 | 0.379 | 23.06 | 0.795 | 0.510 |
| INGP-Base(Müller et al., 2022) | 25.30 | 0.671 | 0.371 | 21.72 | 0.723 | 0.330 | 23.62 | 0.797 | 0.423 |
| INGP-Big(Müller et al., 2022) | 25.59 | 0.699 | 0.331 | 21.92 | 0.745 | 0.305 | 24.96 | 0.817 | 0.390 |
| Mip-NeRF360(Barron et al., 2022) | 27.69 | 0.792 | 0.237 | 22.22 | 0.759 | 0.257 | 29.40 | 0.901 | 0.245 |
| 3D-GS(Kerbl et al., 2023) | 27.21 | 0.815 | 0.214 | 23.14 | 0.841 | 0.183 | 29.41 | 0.903 | 0.243 |
| Mini-GS(Fang & Wang, 2024) | 27.34 | 0.822 | 0.217 | 23.18 | 0.835 | 0.202 | 29.98 | 0.906 | 0.253 |
| Taming-GS(Mallick et al., 2024) | 27.79 | 0.820 | 0.210 | 24.04 | 0.851 | 0.170 | 30.01 | 0.905 | 0.237 |
| Abs-GS(Ye et al., 2024) | 27.47 | 0.819 | 0.192 | 23.66 | 0.852 | 0.162 | 29.61 | 0.901 | 0.236 |
| Pixel-GS (Zhang et al., 2024b) | 27.51 | 0.823 | 0.191 | 23.79 | 0.854 | 0.153 | 28.89 | 0.892 | 0.251 |
| Scaffold-GS(Lu et al., 2024) | 27.67 | 0.813 | 0.223 | 23.96 | 0.853 | 0.177 | 30.21 | 0.906 | 0.255 |
| Perceptual-GS(Zhou & Ni, 2025) | 27.68 | 0.825 | 0.189 | 23.79 | 0.852 | 0.152 | 29.76 | 0.902 | 0.233 |
| Steep-GS(Wang et al., 2025) | 26.99 | 0.793 | 0.249 | 23.43 | 0.836 | 0.194 | 29.56 | 0.899 | 0.254 |
| Ours | 28.01 | 0.826 | 0.189 | 24.40 | 0.860 | 0.150 | 30.04 | 0.907 | 0.243 |

tiveness of our proposed method. Ablation studies demonstrate the effectiveness of our Patch Comparison and Growth Control strategy.

## 4.1. Datases and Implementation Details

**Datasets** We conducted experiments following the dataset settings from 3DGS(Kerbl et al., 2023) on a total of 13 scenes. Specifically, we tested on all 9 scenes from Mip-NeRF360(Barron et al., 2022), two scenes from Tanks&Temples dataset(Knapitsch et al., 2017), and two scenes from Deep Blending(Hedman et al., 2018). The selected scenes represent a variety of capture styles, encompassing both bounded indoor and unbounded outdoor environments. In accordance with the 3DGS protocol, we used every 8th image in each dataset as the test set.

**Implementation** Similar to the original 3DGS, we employed both L1 and SSIM to supervise the optimization of Gaussian attributes. The loss was used to compute the error between the rendered and ground truth images. To ensure stability, edge maps are pre-computed from GT images using a 3x3 high-pass filter. We empirically selected a patch size of $16 \times 16$ as a standard unit for processing. We determined an appropriate budget $B$ based on the final results of the original 3DGS and the Taming-GS. We built our code on the open-source Taming-GS repository. To support our method, we extended the CUDA rasterization framework, enabling the output of additional information, such as the index of the Gaussian contributing most to each pixel. We used the Adam optimizer(Kingma, 2014) for training and implemented our approach in PyTorch(Paszke et al., 2019).

## 4.2. Evaluation

We compare PCGS with Plenoxels(Fridovich-Keil et al., 2022), Instant-NGP(Müller et al., 2022), Mip-NeRF360(Barron et al., 2022), and other 3DGS-based methods(Kerbl et al., 2023; Lu et al., 2024; Ye et al., 2024; Fang & Wang, 2024; Zhang et al., 2024b; Mallick et al., 2024; Zhou & Ni, 2025; Wang et al., 2025). For a fair comparison, all methods were tested on the same machine with an A6000 GPU. When scaling the images for rendering, we used the default scaling factor provided by Mip-NeRF360. For quantitative evaluation, we employed standard metrics: Peak Signal-to-Noise Ratio (PSNR), Structural Similarity Index Measure (SSIM), and Learned Perceptual Image Patch Similarity (LPIPS). In addition to the standard settings, we also provide the points and time of 3DGS, PCGS, and 3DGS-Deblurring Method(Zhang et al., 2024b; Ye et al., 2024; Zhou & Ni, 2025; Fang & Wang, 2024; Mallick et al., 2024; Wang et al., 2025), as shown in Table 3.

*Table 3.* **Prime and Time comparisons on the Mip-NeRF360, Tanks&Temples and Deep Blending Datasets.**

| Dataset | Mip-NeRF360 | | T&T | | Deep Blending | |
|---|---|---|---|---|---|---|
| Method | Time | Primes | Time | Primes | Time | Primes |
| 3DGS(Kerbl et al., 2023) | 42m | 3178K | 27m | 1831K | 36m | 2805K |
| Mini-GS(Fang & Wang, 2024) | 26m | 485K | 16m | 303K | 22m | 557K |
| Abs-GS(Ye et al., 2024) | 33m | 3120K | 15m | 1304K | 37m | 3001K |
| Pixel-GS(Zhang et al., 2024b) | 47m | 5575K | 33m | 4490K | 47m | 4614K |
| Perceptual-GS(Zhou & Ni, 2025) | 47m | 2746K | 26m | 1726K | 51m | 2887K |
| Steep-GS(Wang et al., 2025) | 28m | 2188K | 16m | 1309K | 28m | 1587K |
| Taming-GS(Mallick et al., 2024) | 20m | 3207K | 12m | 1835K | 16m | 2800K |
| Ours | 21m | 2500K | 13m | 1500K | 18m | 1750K |

## 4.3. Results

In this section, we present both quantitative and qualitative evaluations of the proposed method on several real-world scene datasets. As summarized in Table 2, our method consistently outperforms the state-of-the-art (SOTA) on the Mip-NeRF360 and Tanks&Temples dataset and achieves comparable results with the SOTA algorithms on the Deep Blending datasets. Figure 4 presents visual comparisons between our method and SOTA approaches on real-world datasets. These results highlight the effectiveness of our approach in achieving superior reconstruction quality, while accurately capturing fine details in complex scenes.

Furthermore, as shown in Table 3, our method uses fewer Gaussians compared to most existing approaches. Although Mini-GS and Steep-GS employ even fewer Gaussians, they primarily focus on Gaussian simplification, resulting in some degradation in visual quality. Moreover, since our method is built upon Taming-GS—which already accelerates the original 3DGS rendering framework—we mainly compare runtime with Taming-GS, and observe that our method leads to a slight improvement in computation time.

## 4.4. Ablation Studies

We conduct ablation experiments to evaluate the effectiveness of the components of our method. First, we investigate the effect of the patch comparison strategy. Next, we examine the effect of importance score strategy.Finally, we analyze the impact of different budget designs, where we also evaluate a variant without any primitive constraints. This comparison allows us to demonstrate that our performance gains stem from the precision of our targeted densification rather than a simple increase in the total number of Gaussians.

*Table 4.* **Ablation studies on different components of our PCGS.** All metrics are evaluated on the Mip-NeRF360 dataset. Here, PC denotes the proposed Patch Comparison strategy, and IS represents the use of importance scores for Gaussian selection.

| Method | PSNR↑ | SSIM↑ | LPIPS↓ |
|--------|-------|-------|--------|
| FULL   | 28.01 | 0.826 | 0.189  |
| w/o PC | 27.79 | 0.819 | 0.211  |
| w/o IS | 27.84 | 0.822 | 0.203  |

**Patch Comparison** The patch comparison strategy is designed to identify Gaussians that cause artifacts, thereby ensuring their participation in necessary densification. As shown in Table 4 (rows 1 and 2), relying solely on the original view-space position gradients to select candidate Gaussians fails to localize the artifacts, leading to degraded rendering quality. The artifacts illustrated in Figure 6 further highlight the necessity of our approach. Furthermore,

we evaluate the impact of different patch sizes in Table 5. The results demonstrate that our method is robust to various patch scales, with a size of 16 yielding the optimal balance between reconstruction quality and computational efficiency.

*Table 5.* **Ablation studies of different patch size on the Mip-NeRF360 dataset.**

| Patch_size | PSNR↑ | SSIM↑ | LPIPS↓ | Per-iter time |
|------------|-------|-------|--------|---------------|
| 8          | 27.97 | 0.824 | 0.191  | 2.5ms         |
| 16         | 28.01 | 0.826 | 0.189  | 2.3ms         |
| 32         | 27.90 | 0.824 | 0.192  | 2.2ms         |

**Importance Score** The importance score is designed to select appropriate candidate Gaussians for densification. As shown in Table 4 (rows 1 and 3), omitting the importance score and instead selecting candidates uniformly at random leads to a drop in rendering quality. As shown in Figure 7, the edge-based importance scores help our method better reconstruct the structural details in the scene.

**Budget Design** The budget function is designed to control the reasonable growth of the number of Gaussians. We therefore experimented with three different functions—exponential, linear, and logarithmic—to interpolate between the initial number of Gaussians and the final budget. As illustrated in Figure 5, the three curves exhibit distinct growth trends, and since more Gaussians are needed in the early training stages to capture scene details while a stable count is preferred later for refining attributes, the logarithmic function—with its naturally decelerating growth—best aligns with our needs. As shown in Table 6, this interpolation strategy achieves the best results.

To further validate the necessity of this constraint, we also evaluate a configuration that removes the budget limit entirely, performing densification on all identified artifact regions without importance-based filtering. The results, as detailed in Table 7, demonstrate that simply removing constraints leads to a performance degradation despite the increased primitive count. This confirms that our budget-aware strategy is essential, as it prevents redundant primitive explosion and maintains superior reconstruction quality by focusing exclusively on high-impact regions.

*Table 6.* **Ablation studies of the budget designs on the MipNeRF360 dataset.**

| Budget Design | PSNR↑ | SSIM↑ | LPIPS↓ |
|---------------|-------|-------|--------|
| logarithmic   | 28.01 | 0.826 | 0.189  |
| linear        | 27.87 | 0.824 | 0.194  |
| exponential   | 27.84 | 0.821 | 0.198  |

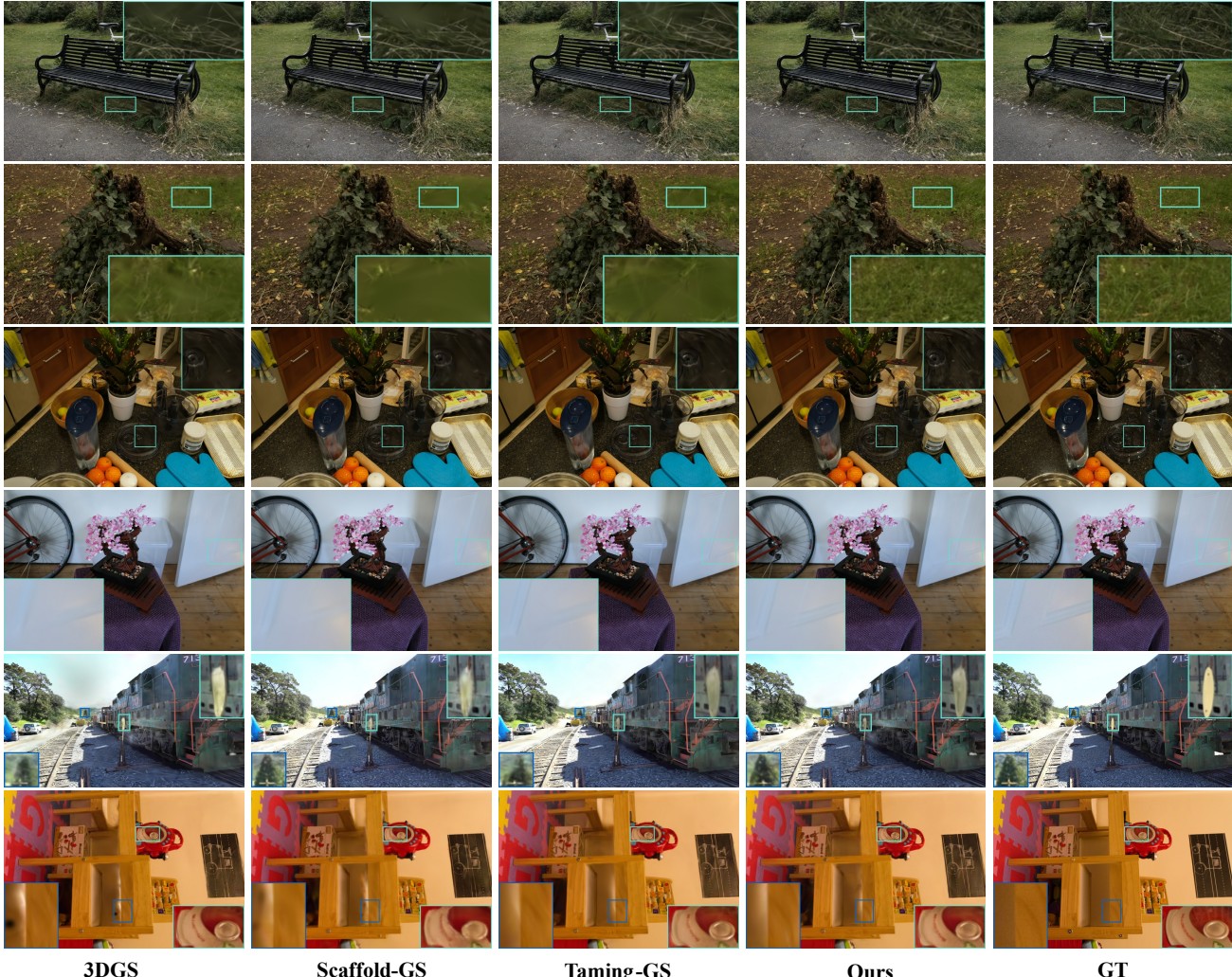

<table>
</table>

|  | 3DGS | Scaffold-GS | Taming-GS | Ours | GT |

*Figure 4.* **Qualitative comparisons of PCGS with three state-of-the-art methods in novel view synthesis.** The comparisons are conducted over multiple indoor and outdoor scenes, including "Bicycle", "Stump", "Counter", and "Bonsai" from Mip-NeRF360, "Train" from Tank&Temple, and "Playroom" from Deep Blending. "GT" denotes the ground-truth images. PCGS demonstrates superior image rendering with fewer artifacts and finer details.

*Table 7.* **Ablation studies of the budget designs on the MipN-eRF360 dataset.**

|  | Dataset | PSNR↑ | SSIM↑ | LPIPS↓ | Primes |
|---|---|---|---|---|---|
| w/o budget | Mip-NeRF360 | 27.88 | 0.825 | 0.196 | 5310K |
|  | T&T | 24.16 | 0.856 | 0.156 | 1675K |
|  | Deep Blending | 29.88 | 0.905 | 0.242 | 2933K |
| w budget | Mip-NeRF360 | 28.01 | 0.826 | 0.189 | 2500K |
|  | T&T | 24.40 | 0.860 | 0.150 | 1500K |
|  | Deep Blending | 30.04 | 0.907 | 0.243 | 1750K |

performs concurrent models both qualitatively and quantitatively. By introducing the Patch Comparison strategy, our method can effectively localize artifacts in the image and guide the densification of Gaussians in artifact regions. Additionally, to mitigate the potential increase in the number of Gaussians caused by the densification process, we design a per-step Gaussian budget framework. We further introduce the importance score to select appropriate Gaussians for densification. The experimental results demonstrate SOTA performance on publicly available benchmark datasets, validating the effectiveness and robustness of our approach.

## 5. Conclusion

In this paper, we propose PCGS, a novel 3D Gaussian splatting (3DGS) approach that focuses on deblurring and out-

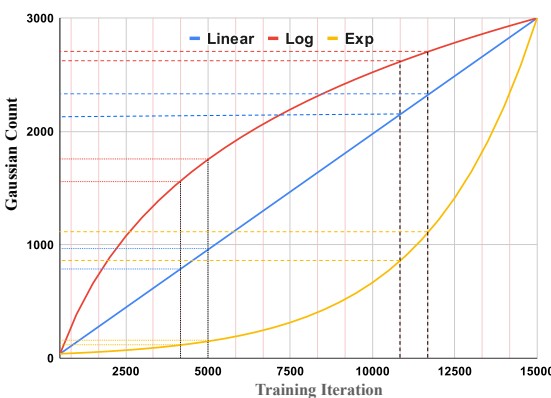

*Figure 5.* **Illustration of different budget functions on the** *flowers* **scene.** The logarithmic growth facilitates rapid Gaussian generation in early stages to encourage geometric exploration, while its subsequent deceleration directs the model's focus toward fine-grained refinement of existing primitives.

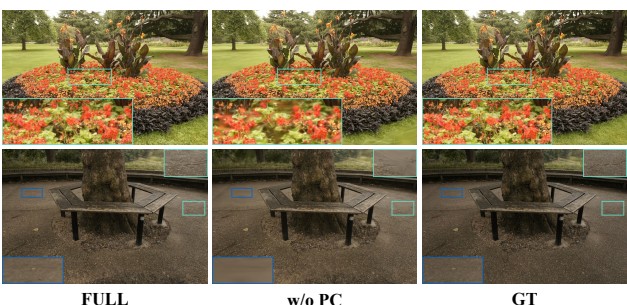

*Figure 6.* **Qualitative comparison for evaluating the effectiveness of Patch Comparison on the** *flowers* **and** *treehill* **scene.**

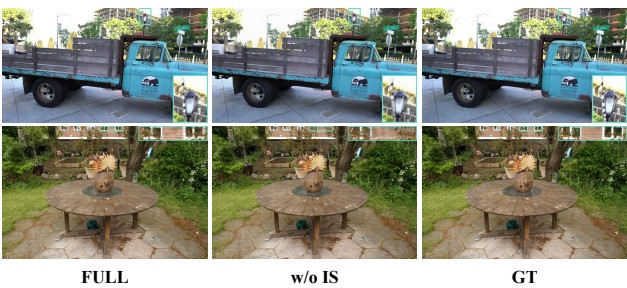

*Figure 7.* **Qualitative comparison for evaluating the effectiveness of Importance Score on the** *truck* **and** *garden* **scene.**

**Limitations** Despite its effectiveness, our method primarily relies on view-dependent optimization, which can occasionally leave complex view-independent artifacts. Crucially, in regions where SfM point clouds are extremely sparse, our explored organic growth may force Gaussians to over-expand in an attempt to fill the geometric vacuum, leading to stretching deformations and artifacts. Future work could further enhance Gaussian localization by inte-

grating auxiliary geometric and perceptual priors, such as depth-based constraints and multi-view consistency. In this context, methods that reinitialize point clouds or introduce new primitives from depth priors would serve as a natural complement to our framework, potentially leading to even more robust scene representations in highly unconstrained environments.

## Acknowledgements

This work was supported by the National Key R&D Program of China (Grant No. 2023YFB3309001).

## Impact Statement

This paper presents work whose goal is to advance the field of Machine Learning. There are many potential societal consequences of our work, none which we feel must be specifically highlighted here.

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

In the appendix, we first describe how we handle the baselines to explain why our reproduced results may differ from those reported in the original papers. We then provide additional details of our method, including a deeper analysis of artifacts and further experimental comparisons.

## A. Baseline Difference

The original 3DGS used all nine scenes from the Mip-NeRF 360 dataset, consisting of five outdoor and four indoor scenes. Its evaluation protocol applied different downsampling factors across scenes—4× for outdoor scenes and 2× for indoor scenes—when generating ground truth images. In contrast, Scaffold-GS (Lu et al., 2024) was not evaluated on the *flowers* and *treehill* scenes and did not perform image downsampling during testing; instead, it followed the default 3DGS setup of resizing high-resolution images to roughly 1600 pixels. Similarly, Steep-GS (Wang et al., 2025) and Abs-GS (Ye et al., 2024) also did not apply any downsampling. To establish a consistent evaluation protocol, we re-ran Scaffold-GS, Steep-GS, and Abs-GS on all nine scenes using the same settings as 3DGS.

Moreover, we observed that the precomputed downsampled images available in the Mip-NeRF 360 dataset (e.g., garden/images_4), generated using ImageMagick, differ noticeably from those obtained using the PIL Image Resize function adopted in the 3DGS codebase. ImageMagick uses a more advanced downscaling algorithm that produces sharper and higher-quality images, making the reconstruction task more challenging. Since the original results of Pixel-GS (Zhang et al., 2024b) and Perceptual-GS (Zhou & Ni, 2025) were obtained using PIL-based resizing, we also re-ran these methods using the precomputed ImageMagick-downscaled images to enable a fairer comparison.

The results of all these re-evaluations are reported in the main paper in Table 2. Additionally, we conducted a comparative experiment on Perceptual-GS using two different downsampling strategies, and the results are reported in Table 8. As shown in the table, the metrics obtained with the PIL Image Resize method are consistently higher than those produced by ImageMagick-based downsampling. For a comprehensive comparison, we also evaluate our method using the PIL-based resizing protocol. On the Mip-NeRF 360 dataset, our method achieves 28.18 PSNR, 0.838 SSIM, and 0.173 LPIPS, consistently outperforming Perceptual-GS under the same settings. These results further demonstrate that our performance gains remain robust across different image-processing protocols.

*Table 8.* **Quantitative results of the two downsampling strategies based on Perceptual-GS on the Mip-NeRF360 dataset.** We observe that using the PIL Image Resize strategy from 3DGS indeed yields a noticeable improvement in performance.

| Method | Metric | stump | bicycle | bonsai | counter | garden | kitchen | room | flowers | treehill | Average |
|---|---|---|---|---|---|---|---|---|---|---|---|
| ImageMagick (-i) | PSNR↑ | 26.97 | 25.48 | 32.50 | 29.28 | 27.48 | 31.67 | 32.01 | 21.44 | 22.26 | 27.68 |
| | SSIM↑ | 0.795 | 0.791 | 0.946 | 0.913 | 0.866 | 0.928 | 0.926 | 0.637 | 0.637 | 0.825 |
| | LPIPS↓ | 0.183 | 0.174 | 0.185 | 0.180 | 0.105 | 0.121 | 0.198 | 0.266 | 0.284 | 0.189 |
| PIL Image Resize (-r) | PSNR↑ | 27.25 | 25.96 | 32.44 | 29.40 | 27.97 | 31.95 | 32.18 | 21.81 | 22.59 | 27.95 |
| | SSIM↑ | 0.806 | 0.805 | 0.952 | 0.921 | 0.877 | 0.935 | 0.935 | 0.654 | 0.656 | 0.838 |
| | LPIPS↓ | 0.177 | 0.165 | 0.152 | 0.157 | 0.099 | 0.107 | 0.169 | 0.257 | 0.274 | 0.173 |

## B. Additional Analysis

**Blur Analysis**    Our key observation is that the traditional Gaussian Splatting training process introduces a substantial imbalance in the contributions of individual Gaussians. In this section, we analyze the underlying causes of blur in 3DGS. Using the *flowers* scene as an example, we trained the original 3DGS and consistently observed noticeable artifacts in the lawn region during both training and testing, as shown in Figure 1. To investigate this issue, we recorded the accumulated view-space positional gradients of the Gaussians that contributed the most to pixels in both well-reconstructed and poorly reconstructed areas from a fixed viewpoint before each densification step. To determine whether any Gaussians in these regions were being selected for densification, we visualized the maximum view-space positional gradients, as shown in Figure 8. We found that in the poorly reconstructed region, the gradients remained consistently below the 2e-4 threshold throughout training. This indicates that **the original 3DGS loss fails to provide meaningful gradients for guiding densification in blurry regions**.

Beyond visualizing the Gaussian with the highest contribution per pixel and the 2D projections of all visible Gaussians in Figure 2, we also recorded how many pixels each Gaussian dominates and summarized the statistics in Figure 9. Specifically,

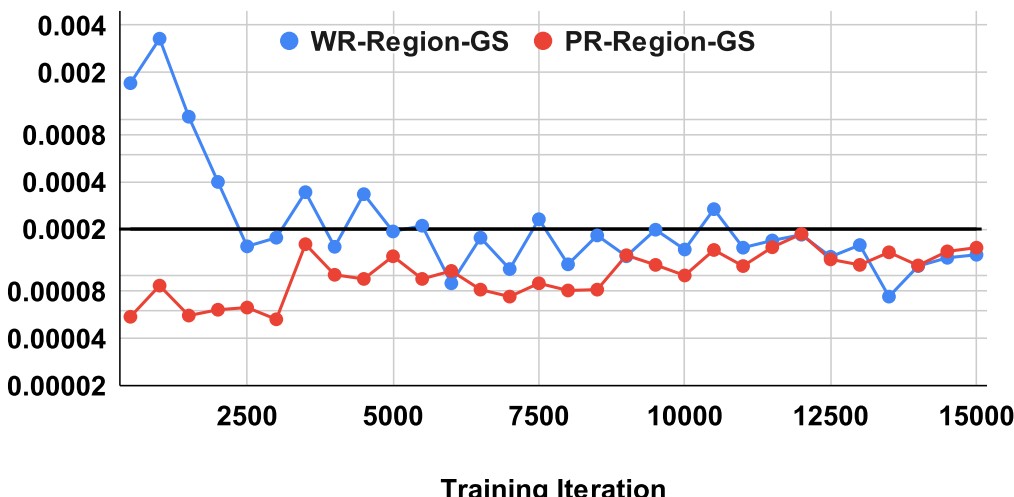

<figure>*Figure 8.* Visualization of maximum view-space positional gradients of the most contributing Gaussians for well-reconstructed (WR) and poorly-reconstructed (PR) regions in the rendered image, using the original 3DGS.</figure>

a small subset of Gaussians often exerts a dominant influence over a large portion of pixels, while other pixels are shaped by contributions from thousands of Gaussians. In regions where Gaussians are sparsely distributed, the rendered image may appear blurry due to insufficient information.

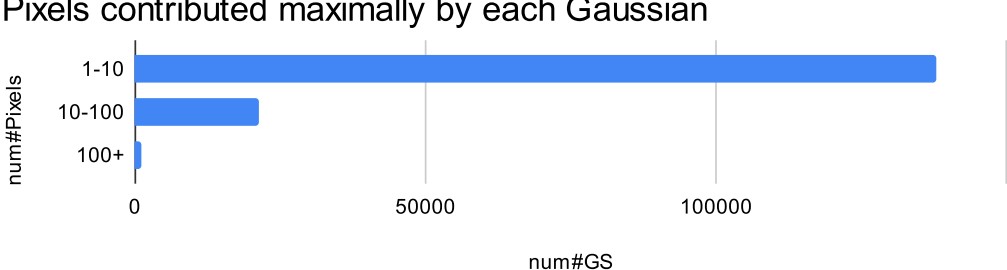

*Figure 9.* Distribution of the number of Gaussians contributing most to each pixel in the rendered image. It can be observed that a large portion of the image is covered by a small number of Gaussians.

To substantiate that our improvements are concentrated in these sparse regions, we conducted a localized evaluation across representative patches: (1) *Well-Reconstructed Patches*, where the baseline already performs well (e.g., the bench in *Bicycle*), and (2) *Under-Reconstructed Patches*, which exhibit significant blurring (e.g., the flowers in *Flowers*). Quantitative results in Table 9 confirm that our method significantly excels in artifact-prone areas, achieving gains of up to +2.26 dB in PSNR and +0.135 in SSIM in under-reconstructed regions. Conversely, well-reconstructed regions show only marginal improvements (avg. 0.5 dB). This disparity proves our adaptive densification precisely targets sparse, high-error regions as hypothesized, enhancing global fidelity by resolving localized structural failures without redundant primitive explosion in stable areas.

Furthermore, regarding the handling of blur in these regions, our Patch Comparison (PC) strategy and the edge map serve distinct yet complementary roles to prevent artifact accumulation in smooth zones. First, these structure-less areas are inherently managed by the standard 3DGS optimization, while our PC strategy further refines their identification by aggregating local artifact pixels, ensuring that contextually important regions are not ignored even if they lack sharp edges. Second, we introduce the Participation Proportion (PP)—defined as the ratio of Gaussians selected by PC to the total densified count—to track this behavioral synergy. As documented in Table 10, the PP values demonstrate that Gaussians responsible for low-frequency artifacts are appropriately selected alongside high-frequency ones. For instance, in edge-rich

datasets like *Deep Blending*, a lower PP aligns with the intuition that less supplementary densification is required in such regions.

**Edge Map**    First, we evaluate the impact of various edge operators on the *Truck* scene. Our framework yields highly consistent metrics across different operators: using our default 3×3 filter, Canny, and Sobel results in PSNR values of 26.10, 26.06, and 26.09. These results demonstrate that our framework is operator-agnostic, ensuring that smooth regions are reconstructed effectively without relying on dense edge maps. Furthermore, to evaluate the computational overhead and time impact introduced by our edge-map pre-computation and patch comparison strategies, we analyze their operational costs on the *Drjohnson* scene. Our approach introduces negligible cost: pre-computing edge maps for all training images takes only 2.06s in total, and the average time spent on patch comparison is merely 0.0023s per iteration. To assess the net impact on total training efficiency, we compared our method with Taming-GS. Once the Gaussian count stabilized (2.0M for ours vs. 3.27M for Taming-GS), our iteration time was 0.0365s compared to 0.0460s for Taming-GS. These results confirm that by maintaining a more compact set of Gaussians, the marginal cost of our refinement logic is more than offset by the gains in overall training speed and reduced memory footprint.

*Table 9.* **Quantitative results on the Well-Rec and Under-Rec patches.**

| Scene | Bicycle | | | | Stump | | | | Flowers | | | |
|---|---|---|---|---|---|---|---|---|---|---|---|---|
| | 3DGS | | Ours | | 3DGS | | Ours | | 3DGS | | Ours | |
| | PSNR↑ | SSIM↑ | PSNR↑ | SSIM↑ | PSNR↑ | SSIM↑ | PSNR↑ | SSIM↑ | PSNR↑ | SSIM↑ | PSNR↑ | SSIM↑ |
| Well-Rec | 26.1579 | 0.8750 | 26.6697 | 0.8805 | 31.9562 | 0.8916 | 32.4850 | 0.9041 | 26.5729 | 0.8565 | 27.0210 | 0.8652 |
| Under-Rec | 25.8944 | 0.6064 | 27.9440 | 0.7953 | 26.4933 | 0.6484 | 27.2782 | 0.6886 | 19.4677 | 0.6417 | 21.7259 | 0.7764 |

**Unmasking Mechanism**    To empirically validate the proposed "unmasking" mechanism, we track the genealogical evolution of Gaussians across successive densification steps. Specifically, each Gaussian is assigned a unique global index upon initialization. During optimization, a splitting Gaussian is replaced by two newly indexed entities, whereas a cloning Gaussian retains its original index while generating a single new one. This lineage tracking enables us to precisely monitor the identity of the dominant (Top-1) alpha-weighted contributor for any given artifact-affected pixel. We then define the Switch Ratio (SR) to quantify the identity transitions of these Top-1 contributors between consecutive densification cycles. Crucially, our analysis isolates the specific Gaussians selected via the Patch Comparison (PC) strategy that ultimately undergo densification. We then evaluate the SR exclusively at the artifact pixels where these targeted Gaussians serve as the primary contributors, comparing identity shifts before and after refinement. Furthermore, we establish a baseline by recording the SR for the same artifact pixels under the standard pipeline without the PC strategy. As illustrated in Table 10, the SR—representing the successful replacement of Gaussians previously dominating artifact regions—increases significantly under our strategy. This transition robustly validates the effectiveness of our "unmasking" mechanism.

*Table 10.* **Quantitative analysis of Gaussian evolution metrics.**

| Scene | PP | SR (3DGS) | SR (Ours) | Split Dom | Clone Dom | Equal | Camera Ext | ER |
|---|---|---|---|---|---|---|---|---|
| Mip-NeRF 360 | 47.96% | 30.86% | 71.16% | 1.90 | 0.07 | 0.12 | 5.16 | 99.40% |
| Tanks&Temples | 40.47% | 38.88% | 75.71% | 1.67 | 0.08 | 0.10 | 6.65 | 99.99% |
| Deep Blending | 22.12% | 40.10% | 72.51% | 0.65 | 0.11 | 0.17 | 7.79 | 99.98% |

## C. Addition Experiments

In this section, we provide additional comparative experiments. Since Pixel-GS produces a significantly different number of Gaussians compared to other baselines, we first adjust its densification threshold to match the Gaussian count of the other methods for a fairer comparison. In addition, because Taming-GS also uses a budget mechanism, we set its budget to be consistent with ours. The corresponding results are reported in Table 11. It can be observed that, with a comparable number of Gaussians, the performance of Pixel-GS decreases on both the Mip-NeRF360 and Tanks&Temples datasets. After setting the budget to match ours, Taming-3DGS shows improvement only in PSNR, while the other metrics decline.

*Table 11.* **Qualitative comparison results across all datasets.** For fairness, Taming-GS is evaluated under the same budget as ours, while Pixel-GS maintains approximately the same number of Gaussians as the original 3DGS.

| Dataset | Mip-NeRF360 | | | | Tanks&Temples | | | | Deep Blending | | | |
|---|---|---|---|---|---|---|---|---|---|---|---|---|
| Method | PSNR↑ | SSIM↑ | LPIPS↓ | Points | PSNR↑ | SSIM↑ | LPIPS↓ | Points | PSNR↑ | SSIM↑ | LPIPS↓ | Points |
| Pixel-GS | 27.44 | 0.816 | 0.212 | 2830K | 23.78 | 0.845 | 0.176 | 1627K | 29.61 | 0.900 | 0.251 | 1937K |
| Taming-GS | 27.85 | 0.818 | 0.213 | 2500K | 24.10 | 0.852 | 0.173 | 1500K | 29.90 | 0.904 | 0.244 | 1750K |
| Ours | 28.01 | 0.826 | 0.189 | 2500K | 24.40 | 0.860 | 0.150 | 1500K | 30.04 | 0.907 | 0.243 | 1750K |

Additionally, qualitative comparisons with Abs-GS (Ye et al., 2024), Perceptual-GS (Zhou & Ni, 2025), Steep-GS (Wang et al., 2025), and Pixel-GS (Zhang et al., 2024b) are shown in Figure 10. Compared to these methods, our approach effectively removes artifacts while preserving the fine details reconstructed by the original 3DGS.

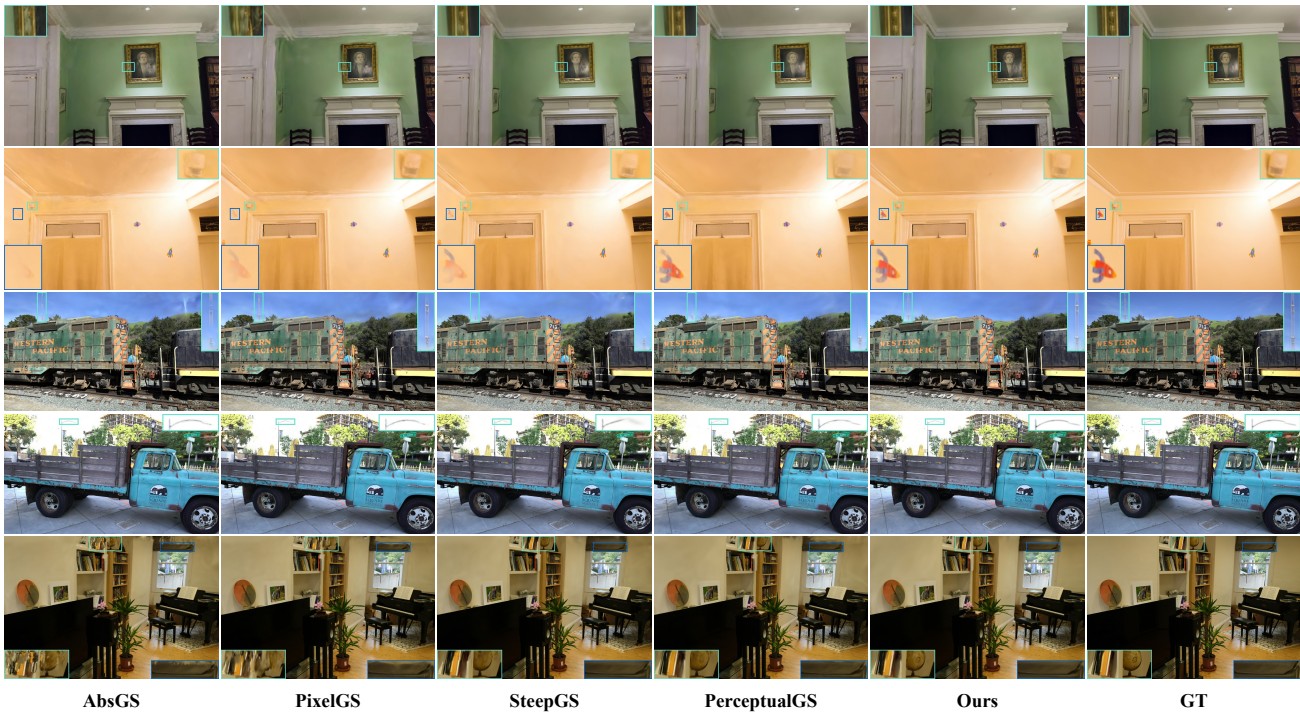

| AbsGS | PixelGS | SteepGS | PerceptualGS | Ours | GT |

*Figure 10.* **Qualitative comparisons of PCGS with four state-of-the-art methods in novel view synthesis.** The comparisons are conducted over multiple indoor and outdoor scenes, including "Drjohnson" and "Playroom" from Deep Blending, "Train" and 'Truck" from Tank&Temple, and "Room" from Mip-NeRF360. "GT" denotes the ground-truth images. PCGS demonstrates superior image rendering with fewer artifacts and finer details.

Regarding the quantifiable limit of organic drift,we conducted an empirical analysis by tracking the spatial displacement of Gaussians. Following (Huang et al., 2025), we defined the initial points obtained from Structure-from-Motion (SfM) as "parent points" and maintained a strict one-to-one lineage correspondence for every newly generated Gaussian. Based on the predominant densification strategy utilized throughout their lineage, Gaussians were categorized as split-dominated (*Split Dom*), clone-dominated (*Clone Dom*), or equal. To quantify this drift, we recorded the average Euclidean distance between the final positions of the Gaussians and the initial coordinates of their respective parent points, using the camera extent—defined as $1.1\times$ the radius of the smallest bounding sphere encompassing all camera positions—as a standardized reference for the scene's spatial scale. Furthermore, we quantified spatial exploration by voxelizing the Gaussian positions prior to densification, using the median initial point distance as the voxel size. During splitting, we recorded the exploration ratio (*ER*)—the proportion of newly split Gaussians migrating into previously unoccupied voxels.

The results in Table 10 indicate that during densification, the split mechanism fundamentally drives spatial exploration by encouraging Gaussians to migrate into unpopulated regions, whereas the clone mechanism primarily focuses on localized

refinement. The extent of this exploration varies distinctly across scene types: in 360-degree scenes (*Mip-NeRF 360* and *Tanks & Temples*), Gaussians can "organically" drift significantly into unpopulated areas due to dense multi-view constraints. In contrast, for exploratory trajectories (*Deep Blending*), the effective range for Gaussian movement remains considerably more restricted.

