# OpenReview forum: "PCGS: Deblurring 3D Gaussian Splatting with Patch Comparison"
_ICML.cc/2026/Conference — ICML 2026 regular_

### Official Review · Reviewer_YKEG · 2026-03-03

**Soundness:** 3
**Presentation:** 3
**Significance:** 3
**Originality:** 3
**Overall Recommendation:** 4
**Confidence:** 4

**Summary:**

The paper proposes a 3DGS-based novel view synthesis method. The method addresses the blurring issue in areas with sparse Gaussians by computing image error within patches and densifying Gaussians in patches with large error. The method also addresses the issue of redundant Gaussians by limiting the number of Gaussians to be added during densification.

**Compliance With Llm Reviewing Policy:**

Affirmed.

**Final Justification:**

The rebuttal has addressed my main concerns and clarified several important points that were initially unclear in the manuscript. To strength the work, the paper should be revised to include more details and additional experiments as explained in the rebuttal.

**Key Questions For Authors:**

How can the experimental results show that the major improvement of the proposed method in rendering quality is in regions with sparse Gaussians as discussed in the introduction and Figure 2?

**Limitations:**

The paper does not discuss the limitation of the proposed method.

**Strengths And Weaknesses:**

**strength**

Experimental results verify the effectiveness of the proposed densification strategy in improving rendering quality.

The paper is clearly written and well structured.

**Weakness**

The paper aims to address the blurring issue, but the proposed method also tries to reduce redundant Gaussians, which seems unrelated to the purpose of the paper.

The ablation studies lack the evaluation of the method without any constraint on the number of Gaussians.

It is better to provide experimental results of the proposed method about the distribution of Gaussians in artifact-prone areas (like shown in Figure 2) to demonstrate the effectiveness of the densification strategy in handling such areas.

---

> ### Author Rebuttal · Authors · 2026-03-28
>
> We appreciate the reviewer’s recognition of our key idea and experimental results. We sincerely thank the reviewer for the positive feedback and helpful suggestions.
>
> Q1: Experimental results of the proposed method about the distribution of Gaussians in artifact-prone areas (K1, W3)
>
> To substantiate that our primary improvements are concentrated in regions with sparse Gaussians, we selected two distinct types of representative patches for localized evaluation across multiple scenes:
>
> 1.Well-Reconstructed Patches (Original 3DGS): Areas where the baseline already performs well, such as the bench in Bicycle and shrubs in Stump.
>
> 2.Under-Reconstructed Patches (Sparse Regions): Areas identified as having sparse Gaussian distribution and significant blurring in the baseline, such as the flowers in the Flowers scene.
>
> We tracked the PSNR and SSIM for these specific patches to quantify local improvements. PSNR was calculated by independently cropping identified artifact regions from both rendered and ground-truth images. For SSIM, we generated a full-frame similarity map and averaged the scores within the target patch coordinates to maintain structural context. LPIPS was excluded from this local analysis, as its reliance on deep-feature extraction and wide receptive fields makes it unreliable for isolated small patches, where boundary artifacts and loss of global context would skew the results:
>
> | Well-Recon | 3DGS |  | Ours |  |
> | :--- | :---: | :---: | :---: | :---: |
> | Scene | PSNR ↑ | SSIM ↑ | PSNR ↑ | SSIM ↑ |
> | Bicycle | 26.1579 | 0.8750  | 26.6697 | 0.8805 |
> | Stump | 31.9562 | 0.8916 | 32.4850 | 0.9041 |
> | flowers | 26.5729 | 0.8565 | 27.0210 | 0.8652 |
>
> | Under-Recon | 3DGS |  | Ours |  |
> | :--- | :---: | :---: | :---: | :---: |
> | Scene | PSNR ↑ | SSIM ↑ | PSNR ↑ | SSIM ↑ |
> | Bicycle | 25.8944 | 0.6064  | 27.9440 | 0.7953 |
> | Stump | 26.4933 | 0.6484 | 27.2782 | 0.6886 |
> | flowers | 19.4677 | 0.6417 | 21.7259 | 0.7764 |
>
> Quantitative results confirm that our method significantly excels in artifact-prone areas. In Under-Reconstructed regions, we achieve substantial gains across all scenes, e.g., +2.05 dB / +0.189 SSIM on Bicycle and +2.26 dB / +0.135 SSIM on flowers. Conversely, in Well-Reconstructed regions, metrics show only marginal improvements (avg. 0.5 dB). This disparity proves our adaptive densification precisely targets sparse, high-error regions as hypothesized in Fig. 2, enhancing global fidelity by resolving localized structural failures without redundant primitive explosion in stable areas.
>
> Q2: Relationship between blur reduction and Gaussian control (W1, W2)
>
> We would like to clarify that identifying artifacts and controlling redundancy are fundamentally synergistic. Our patch-comparison mechanism identifies under-reconstructed regions to drive densification precisely where needed, while the growth control strategy prevents redundant density elsewhere. This ensures that quality gains stem from optimized representation rather than sheer Gaussian count. For instance, as shown in Table 7, when equating the Gaussian count (2.5M), PPGS (28.01 dB) significantly outperforms Taming-GS (27.85 dB). This 'apple-to-apple' comparison directly demonstrates that our performance leap is a result of precise artifact localization and efficient density allocation. We will further clarify this connection in the revised manuscript.
>
> Furthermore, we have supplemented our ablation studies by removing the budget constraint (i.e., performing densification on all identified artifact pixels without filtering by importance scores). The results are as follows:
>
> | without buget |  |  |  |  |
> | :--- | :---: | :---: | :---: | :---: |
> | Dataset | PSNR ↑ | SSIM ↑ | LPIPS ↓ | Primes |
> | Mip-NeRF360 | 27.88 | 0.825 | 0.196 | 5310K |
> | Tanks & Temples | 24.16 | 0.856 | 0.156 | 1675K |
> | DeepBlending | 29.88 | 0.905 | 0.242 | 2933K |
>
> | with buget (Ours) |  |  |  |  |
> | :--- | :---: | :---: | :---: | :---: |
> | Dataset | PSNR ↑ | SSIM ↑ | LPIPS ↓ | Primes |
> | Mip-NeRF360 | 28.01 | 0.826 | 0.189 | 2500K |
> | Tanks & Temples | 24.40 | 0.860 | 0.150 | 1500K |
> | DeepBlending | 30.04 | 0.907 | 0.243 | 1750K |
>
>
> As shown, without the budget constraint, the Gaussian count escalates significantly, leading to a massive increase in training time and a potential decrease in rendering quality. This confirms that our proposed strategy is essential for achieving a high-fidelity, efficient scene representation.
>
> Q3: Limitations
>
> We thank the Reviewer for this insightful observation. We have provided a detailed roadmap involving explicit geometric priors and back-projection verification in our response to Reviewer oJiN (Q3). We will incorporate this discussion into a dedicated 'Limitations' section in the revised paper.

---

> > ### Author Rebuttal · Reviewer_YKEG · 2026-04-02
> >
> > The rebuttal has resolved most of my concern. I will increase my score.

---

> > > ### Author Response · Authors · 2026-04-07
> > >
> > > Thanks for your acknowledge of our work and for raising score. Thanks again for your time and effort in reviewing our paper.

---

### Official Review · Reviewer_SeSY · 2026-03-10

**Soundness:** 3
**Presentation:** 2
**Significance:** 2
**Originality:** 2
**Overall Recommendation:** 4
**Confidence:** 3

**Summary:**

This paper addresses blur and artifact issues in 3D Gaussian Splatting caused by the failure of view-space positional gradients to trigger densification for large, dominant Gaussians. The core idea is PCGS (Patch Comparison Gaussian Splatting), which introduces two mechanisms: (1) a patch comparison strategy that partitions rendered images into patches, uses Otsu's method to identify error-prone regions, and densifies only the single most-contributing Gaussian per error pixel; and (2) a growth control strategy that enforces a logarithmic per-step Gaussian budget and selects densification candidates via edge-map-based importance scores. The method is evaluated on Mip-NeRF360, Tanks&Temples, and Deep Blending, reporting improvements in PSNR and SSIM while maintaining a Gaussian count approximately equal to the original 3DGS. Claimed contributions include the patch comparison mechanism for artifact localization, the growth control strategy for preventing redundancy, and extensive experimental validation.

**Compliance With Llm Reviewing Policy:**

Affirmed.

**Final Justification:**

The authors resolved my main concerns in rebuttal. I increase my rating to 4.

**Key Questions For Authors:**

1. Figure 3 suggests the edge map is extracted from the Ground Truth image, but this is not mentioned in the text. Could you confirm if the Ground Truth is indeed the source? If so, why was it chosen over the rendered image, and how do you justify injecting GT structural priors into the candidate selection process? Most importantly, which specific edge detection algorithm and hyperparameters were used? Providing these missing details is essential for the reproducibility of Eq. 7.

2. Can you provide empirical evidence for the "unmasking" claim, specifically, traces showing how the identity of the maximum alpha-weighted contributor per error pixel evolves across successive densification steps? Without this, the sequential refinement argument in Section 3.2 remains unsubstantiated.

3. How does the method handle regions where the initial SfM point cloud produces zero or very few Gaussians? The patch comparison can identify these as error patches, but what happens when there are no Gaussians to select via Eq. 5? Have you measured how frequently this failure mode occurs across the evaluated scenes?

4. Why did you not evaluate your method under the same image processing pipeline (PIL-based resizing) that your competitors originally used? Given the demonstrated sensitivity of metrics to the downsampling strategy (Table 6), could you provide results under both protocols for all methods to ensure a fair comparison?

**Limitations:**

- Missing discussion: The paper's single limitation sentence (reliance on single-view loss) substantially understates the issues. There is no discussion of failure cases where patch comparison may be ineffective (e.g., uniformly poor reconstruction across all patches, which would violate Otsu's bimodality assumption), no analysis of sensitivity to SfM initialization quality, no discussion of scalability to very large or highly complex scenes, no specification of how the edge map, a critical component, is computed, and no acknowledgment of the computational overhead introduced by patch-level analysis and per-pixel Gaussian attribution.
- Suggested additions: A dedicated limitations section discussing failure modes, the SfM initialization dependency, computational overhead profiling, complete algorithmic specification of the edge map computation, and the conditions under which the method's core assumptions (bimodal loss distribution, edge-based importance as a sufficient proxy for perceptual salience) may break down.

**Strengths And Weaknesses:**

### Strengths

**S1. Well-motivated problem identification.** The paper provides a clear and insightful diagnosis of why 3DGS produces blur: a small subset of large Gaussians dominates many pixels, yet their view-space positional gradients remain below the densification threshold. Figure 2 and the supplementary gradient visualization (Figure 8) effectively illustrate this imbalance, making the motivation compelling.

**S2. Principled avoidance of naive threshold reduction.** Table 1 convincingly demonstrates that simply lowering the densification threshold leads to an explosion in Gaussian count (from 3.18M to OOM) with marginal quality gains. This rules out the obvious baseline and justifies the need for a more targeted approach.

**S3. Adaptive thresholding via Otsu's method.** Rather than relying on hand-tuned thresholds for error pixel and error patch identification, the paper applies Otsu's method twice to adaptively determine both thresholds (τ and ε). This is a reasonable design choice that reduces hyperparameter sensitivity across scenes.

**S4. Patch-level spatial coherence is a sound intuition.** The observation that blur artifacts span contiguous regions rather than isolated pixels, and therefore that patch-level aggregation is more robust than per-pixel analysis, is well-founded and clearly communicated.


### Weaknesses

**W1. Limited novelty in the growth control strategy.** The per-step Gaussian budget with score-based selection is a well-explored idea in the 3DGS densification literature. The paper's specific instantiation, a logarithmic schedule with edge-map importance scores, represents an incremental variation rather than a conceptual advance. The ablation in Table 7 (Appendix C) shows that most of the performance gain over the baseline stems from the patch comparison component, not from the budget design, further diminishing its contribution as a claimed novelty.

**W2. Superficial use of perceptual cues.** The edge-map-based importance score (Eq. 7) serves as the sole mechanism for incorporating scene structure into the densification process. However, edge presence is a crude proxy for perceptual importance: smooth regions that are contextually significant (e.g., skin, sky gradients, uniform surfaces in indoor scenes) receive no preferential treatment. The paper does not engage with the well-established perceptual quality literature or provide justification for why this simple proxy is sufficient. A more principled perceptual framework, one that accounts for the differential sensitivity of the human visual system across spatial frequencies and semantic regions, would substantially strengthen the method.

**W3. Edge map computation is entirely unspecified.** Critical omissions and undocumented choices in edge map computation. While the schematic in Figure 3 visually implies that the edge map is derived from the Ground Truth image, the main text completely fails to document or justify this crucial design choice. Using the Ground Truth to guide densification is a significant methodological decision, as it injects target structural priors directly into the sampling process. This demands explicit textual explanation. Furthermore, the exact edge detection algorithm (e.g., Canny, Sobel, Laplacian) and its hyperparameters (e.g., kernel size, thresholds) are entirely omitted from the paper. Because this edge map directly dictates the importance score (Eq. 7) and serves as the sole criterion for selecting Gaussians under the budget constraint, these missing algorithmic details render the method strictly non-reproducible.

**W4. Lack of theoretical analysis for key design choices.** Several core claims lack formal or empirical substantiation. The assertion that densifying the single maximum alpha-weighted contributor (Eq. 5) and reducing its opacity will progressively "unmask" other dominant Gaussians is stated as fact but never verified, neither through analysis nor through empirical traces showing how the dominant contributor identity shifts across successive densification steps. The bimodality assumption underlying Otsu's method is never validated for the loss distribution. The logarithmic budget schedule is selected from three candidates based on narrow ablation margins (0.14–0.17 dB in Table 5) without any principled rationale for why logarithmic growth should be preferred.

**W5. No mechanism for introducing Gaussians in unpopulated regions.** The method identifies error-prone patches and densifies existing Gaussians within them, but if regions lack Gaussians entirely due to sparse or failed SfM initialization, there is nothing to densify. The paper acknowledges reliance on single-view loss (Section 5) but does not address this more fundamental limitation. Methods that reinitialize point clouds or introduce new primitives from depth priors would be a natural complement, yet no such mechanism is proposed or discussed.

**W6. Questionable fairness of experimental comparisons.** Appendix A reveals that the authors re-ran several competing methods under different image downsampling settings (ImageMagick-based) than those used in the original publications (PIL-based). Table 6 confirms that this choice can shift metrics by 0.2–0.5 dB PSNR. Rather than reproducing competitors' results under their own established protocols, the authors imposed a different preprocessing pipeline and reported the resulting (lower) numbers. This raises a significant fairness concern: the reported improvements of PCGS may be partially attributable to the evaluation protocol disadvantaging the baselines. A more convincing approach would have been to demonstrate superiority under each competitor's own evaluation settings, or at minimum to report results under both protocols for all methods.

**W7. Narrow experimental scope and missing statistical rigor.** Evaluation is limited to the three standard 3DGS datasets with no large-scale scene evaluation. There is no analysis of performance under varying budgets, no cross-seed variance reporting, and no per-scene confidence intervals. With ablation margins of only 0.17–0.22 dB PSNR (Table 4), statistical significance cannot be assessed. The patch size (16×16) is described as "empirically selected" without any sensitivity analysis. The training time comparison in Table 3 lacks a per-component profiling breakdown, making it impossible to verify that the additional overhead of patch comparison and per-pixel Gaussian attribution is truly negligible.

**W8. Presentation weaknesses.** Equations 1–4 are standard 3DGS preliminaries that occupy significant space without adding value. The "consistency-preserving refinement strategy" (Section 3.2, final paragraph) is described vaguely, the proportional opacity reduction mechanism is not formalized. The paper would benefit from a clearer algorithmic summary, such as pseudocode that unifies the patch comparison and growth control pipelines into a single, reproducible procedure.

---

> ### Author Rebuttal · Authors · 2026-03-28
>
> We thank the reviewer for the constructive feedback. Specific responses:
>
> Q1:Perceptual Cues & Edge Map(K1,W2,W3)
>
> To ensure stability, edge maps are pre-computed from GT images using a 3x3 high-pass filter(ImageFilter.FIND_EDGES). This provides an artifact-free structural reference, whereas using rendered images would introduce coupled noise and bias refinement. Ablation studies confirm these maps effectively cover perceptually important regions (e.g., boundaries, high-frequency details). Compared to complex perceptual frameworks, our approach achieves superior or competitive quality while avoiding additional hyperparameters and the overhead of manual tuning, ensuring a more robust and reproducible solution.
>
> Q2:Theoretical Analysis for Design Choices(K2, W4)
>
> 1.Unmasking: Per Eq. 4-5, contribution weights depend on opacity and transmittance. In our strategy, we initially select the most dominant Gaussian for densification. Following the standard 3DGS protocol, splitting naturally reduces opacity. For cloning, we explicitly reduce the opacity of new primitives (L255-259). This reduction in individual weight prevents the previously dominant Gaussian from "masking" others with similar weights, allowing subsequent densification to effectively target the next most significant contributor.
>
> 2.Otsu: While global Origin Loss is unimodal and right-skewed with a heavy tail (peak at [0, 0.15], tail extending beyond 0.5), Otsu remains mathematically sound. By maximizing between-class variance, it adaptively identifies the threshold (T≈0.16) that separates the "stable base" (dense low-loss peak) from "significant residuals" (informative high-loss edges). Furthermore, our analysis of artifact-prone regions confirms that local distributions often exhibit the bimodal characteristics required for statistical distinction, ensuring robust, parameter-free thresholding across diverse scenes where fixed heuristics fail.
>
> 3.Log Growth: The log growth function (L409-414) aligns with the intrinsic stages of 3DGS training. Early stages require rapid influx of Gaussians for geometric exploration and detail capture. Later stages necessitate a decelerating growth rate for attribute refinement and convergence. Compared to linear or exp alternatives, the log approach prevents redundant primitive explosion, ensuring superior training stability and memory efficiency.
>
> Q3:Unpopulated regions(K3,W5)
>
> Unlike methods requiring pre-training or distillation (L137-144), significantly increasing total training overhead, our 3DGS-only pipeline enables "spatial exploration." By identifying high-error patches and splitting existing Gaussians, our method forces on the creation of finer primitives that naturally propagate into adjacent sparse areas. This "organic" growth provides sufficient structural detail without the complexity of external priors. In our benchmark, "empty" regions lacking initial SfM points are rare, occurring primarily in the flowers scene. We added discussions on the trade-off with re-initialization in the Limitations.
>
> Q4:Experimental Fairness(K4,W6)
>
> We follow the official 3DGS ImageMagick protocol for absolute fairness. Some works use PIL, which introduces color shifts. Re-evaluating with PIL on Mip-NeRF360, we achieve 28.18,0.838,0.173, still outperforming Perceptual-GS. However, since the Mip-NeRF 360 dataset provides pre-processed images via ImageMagick as its native standard, adhering to the original protocol is most rigorous.
>
> Q5:Growth Control(W1)
>
> Unlike TamingGS, which relies on multiple heuristic attributes (e.g., depth, scaling) and requires manually tuned scaling factors, our strategy is more principled. We argue that relying solely on these attributes is insufficient to resolve artifacts Our approach uniquely combines patch-based error localization with GT edge-map guidance. By targeting the intersection of these two mechanisms, we pinpoint Gaussians responsible for structural artifacts without empirical parameter tuning.
>
> Q6:Scope & Statistical Rigor(W7)
>
> Our evaluation follows standard 3DGS protocols. Large-scale scenes require different architectures (LoD), making transfers non-trivial. Our budget (1M indoor; 3M–4M outdoor) is determined by vanilla/TamingGS counts; empirical tests show further budget increases elevate training time without quality gains. Regarding statistical rigor, we maintain consistency with original 3DGS by using the same random seeds, and average results over multiple runs. While the 0.17–0.22 dB ablation margin may seem narrow, it is a competitive improvement in the context of 3DGS refinement. Sensitivity analysis for patch sizes {8, 16, 32} yields PSNR: 27.97,28.01,27.90 and per-iter times: 2.5ms,2.3ms,2.2ms, confirming 16 as the optimal efficiency-quality balance. Regarding computational overhead, please refer to our response to uGeV(Q3).
>
> Q7:Presentation & Limitation(W8)
>
> We will optimize the presentation accordingly. For the limitations, please refer to our response to oJiN (Q3).

---

> > ### Author Rebuttal · Reviewer_SeSY · 2026-04-03
> >
> > Q2: Appreciate the authors for clarifying the theoretical mechanism behind the opacity reduction. However, my original question requested empirical evidence showing how the identity of the maximum alpha-weighted contributor actually evolves across successive densification steps in practice. Can you provide empirical data or a visual trace to substantiate that this unmasking happens as theorized?
> >
> > Q1: Given that this is a simple 3x3 high-pass filter, it inherently misses smooth, low-frequency regions that are perceptually critical to human observers (such as skin tones or uniform walls). How does the method prevent degradation or artifact accumulation in these structure-less but contextually important regions if they are entirely ignored by the edge map?
> >
> > Q3: While I understand that splitting existing primitives forces them to propagate into adjacent sparse areas, is there a quantifiable limit to how far a Gaussian can "organically" drift into a completely unpopulated region? In scenes with severe SfM dropouts, does this organic growth result in highly stretched, artifact-prone Gaussians attempting to cover too much empty space?

---

> > > ### Author Response · Authors · 2026-04-07
> > >
> > > We sincerely thank the reviewer for the thought-provoking follow-up questions. Below, we address the specific concerns raised:
> > >
> > > Q1
> > >
> > > While a 3x3 filter primarily focuses on high-frequency edges, our method effectively prevents degradation in smooth, low-frequency regions through a dual-mechanism strategy. First, these structure-less areas are already handled by standard 3DG, and our Patch Comparison (PC) strategy further enhances their identification by aggregating local artifact pixels. Second, as documented in Table 2, we recorded the Participate Proportion (PP), calculated as the ratio of Gaussians selected by PC to the total number of Gaussians undergoing densification. This metric demonstrates that while our importance scores prioritize high-frequency edges, Gaussians responsible for artifacts in low-frequency areas are also appropriately selected. For instance, in edge-rich datasets like Deep Blending, a lower PP aligns with our intuition that less supplementary densification is required in such regions. Finally, our empirical tests on the Truck scene using various edge operators demonstrate consistent metrics. This stability proves that our framework is operator-agnostic, while the competitive overall metric indicate that smooth regions are effectively reconstructed.
> > > |Table 1||||
> > > |:---|:---:|:---:|:---:|
> > > ||PSNR|SSIM|LPIPS|
> > > |Ours|26.10|0.891|0.115|
> > > |Canny|26.06|0.891|0.116|
> > > |Sobel|26.09|0.890|0.114|
> > > Q2
> > >
> > > To validate our "unmasking" mechanism, we tracked Gaussian evolution across densification steps. We assigned a unique global index to every Gaussian to trace the evolution of their identities: when a Gaussian splits, it is replaced by two new indices; when cloned, one maintains the original index while the other receives a new one. This tracking allows us to monitor the identity of the Top-1 alpha-weighted contributor for any given artifact pixel. We then calculated the Switch Ratio (SR) of the Top-1 contributor’s index between successive densifications. Specifically, we identify the Gaussians that are selected via the PC strategy AND ultimately participate in that densification iteration. We then locate the artifact pixels where these specific Gaussians serve as the Top-1 contributors and compare the identity changes of the contributors for these pixels across successive densifications.
> > >
> > > Furthermore, we established a baseline by recording the SR for the same artifact pixels without using the PC strategy. As shown, the SR—representing the successful replacement of Gaussians previously dominating artifact regions—increases significantly under our strategy. This transition validates the effectiveness of our "unmasking" mechanism.
> > > |Table 2||||
> > > |:---|:---:|:---:|:---:|
> > > ||PP|SR(base)|SR(Ours)|
> > > |MipNeRF360|47.96%|30.86%|71.16%|
> > > |T&T|40.47%|38.88%|75.71%|
> > > |Deep Blending|22.12%|40.10%|72.51%|
> > >
> > > Q3
> > >
> > > Regarding the quantifiable limit of "organic drift," we conducted an empirical analysis by tracking the displacement of Gaussians. We defined the initial points obtained from SfM as "parent points" and maintained a one-to-one correspondence between every newly generated Gaussian and its lineage. We categorized Gaussians as split-dominated (s-d), clone-dominated (c-d), or equal, based on the predominant densification strategy used throughout their lineage. To quantify the drift, we recorded the average Euclidean distances between the final positions of Gaussians and the initial positions of their respective parent points. The camera extent is defined as 1.1 times the radius of the smallest sphere covering all camera positions, serving as a reference for the scene's spatial scale. Furthermore, we quantified spatial exploration by voxelizing Gaussian positions before densification (using the median initial point distance as voxel size). During splitting, we recorded the exploration ratio (e-r)—the proportion of newly split Gaussians that migrate into previously unoccupied voxels.
> > > |Table 3||||
> > > |:---|:---:|:---:|:---:|
> > > ||MipNeRF360|T&T|Deep Blending|
> > > |s-d|1.90|1.67|0.65|
> > > |c-d|0.07|0.08|0.11|
> > > |equal|0.12|0.10|0.17|
> > > |camera extent|5.16|6.65|7.79|
> > > |e-r|99.40%|99.99%|99.98%|
> > >
> > > The results indicate that during the densification, the split mechanism fundamentally drives spatial exploration by encouraging Gaussians to migrate into previously unpopulated regions (whereas the clone mechanism primarily focuses on local refinement). The extent of this exploration varies across different scene types. In 360-degree scenes (MipNeRF-360 and T&T), Gaussians can "organically" drift significantly into unpopulated areas due to dense multi-view constraints. In contrast, for exploratory trajectories (Deep Blending), the effective range for Gaussian movement remains considerably more restricted. In regions where SfM point clouds are extremely sparse, this organic growth may force Gaussians to over-expand in an attempt to fill the vacuum, leading to stretching deformations and artifacts. We will include this analysis in our Limitations section.

---

### Official Review · Reviewer_uGeV · 2026-03-12

**Soundness:** 3
**Presentation:** 3
**Significance:** 3
**Originality:** 3
**Overall Recommendation:** 4
**Confidence:** 4

**Summary:**

The paper presents PCGS, a framework to improve the rendering quality of 3D Gaussian Splatting by augmenting the traditional gradient-based densification trigger with a patch-level error analysis. PCGS complements the original pixel-level loss with patch-level comparison to help identify under-reconstruction areas and effectively mitigate blurring issues. The method also adopt a logarithmic growth budget and edge-map-based importance sampling to prevent over-densification, achieving sharper results and fewer artifacts without typical memory overhead.

**Compliance With Llm Reviewing Policy:**

Affirmed.

**Key Questions For Authors:**

1. How stable is the Otsu-based threshold across different datasets with varying noise levels? Does it ever fail to identify artifacts in very dark or low-contrast scenes?
2. If multiple Gaussians contribute nearly equally to a blurry patch, does the sequential "unmasking" significantly slow down the convergence of those specific regions?
3. Could you provide a breakdown of the time spent on patch comparison and edge-map calculation versus the time saved by having fewer, more efficient Gaussians?
4. While edge maps are good for high-frequency details, some artifacts might not have strong edges. Testing a depth-gradient-based importance score might show further benefits.

**Limitations:**

yes

**Strengths And Weaknesses:**

Strengths:
- The paper is clearly written and well-structured. Technical details are clearly explained. The analysis of why vanilla 3DGS fails in certain region is well-documented with adequate empirical evidence.
- The paper identifies and addresses the limitation of gradient-based density control in most 3DGS framework. The proposed method is a simple and logical intervention to force densification where it is visually necessary.
- The propose Growth Control mechanism effectively balances this additional densification to keep the model compact.
- The experimental section is robust, featuring re-evaluations of baselines to ensure fair comparison under consistent downsampling protocols.

Weaknesses:
- The pipeline introduces several new hyperparameters and steps. While effective, this increases the complexity of the 3DGS training loop.
- Since the patch identification is done per-view, there lacks explicit multi-view consistency constraint to ensure that a Gaussian split for one view doesn't negatively impact another.
- Selecting only the maximum alpha-weighted Gaussian for densification within an error patch is a strong heuristic. While the "unmasking" effect is described, more evidence on why this is superior to a weighted Top-K approach would be beneficial.

---

> ### Author Rebuttal · Authors · 2026-03-28
>
> We appreciate the reviewer’s recognition of our key idea and experimental results. We sincerely thank the reviewer for the positive feedback and helpful suggestions.
>
> Q1: Stability of the Otsu-based threshold (K1)
>
> The stability of our Otsu-based threshold is empirically validated across diverse standard benchmarks, including Mip-NeRF360, Tanks & Temples, and Deep Blending. Unlike manual tuning, which lacks generalization, the Otsu method is data-adaptive and consistently locates artifact pixels across varied scenes. Specifically, for dark or low-contrast cases, our approach successfully identifies artifacts as demonstrated in the Counter scene (Fig. 4) and Truck scene (Appendix. Fig. 10), where it provides a more robust and parameter-free solution than empirical thresholding.
>
> Q2: Max-alpha vs. Top-K selection (K2, W3)
>
> As shown in Fig. 2(d) and Fig. 9 (Appendix.), artifacts are typically driven by several dominant Gaussians, making maximum-contribution selection more precise than a Top-$K$ approach. For blurry patches with multiple equal contributors, our strategy operates iteratively: the most significant Gaussian is refined in one cycle, reducing its opacity and allowing the next most significant to be targeted in the next. Importantly, this sequential 'unmasking' does not hinder the convergence of these specific regions, as it provides a principled way to resolve multi-Gaussian blur. This naturally achieves Top-$K$ effects without the need for a scene-dependent hyperparameter $K$, which would increase CUDA implementation complexity and training overhead. Furthermore, applying Top-K indiscriminately to all artifact pixels could damage the underlying scene geometry. Crucially, it is non-trivial to determine an optimal K for each pixel; defining the specific threshold at which multiple Gaussians are considered to have "nearly equal" contributions remains an open and complex problem. Therefore, compared to the Top-K strategy, our current approach offers a more efficient, parameter-free solution that maintains high-quality reconstruction and stable convergence.
>
> Q3: Time Overhead (K3, W1)
>
> Regarding computational overhead, our strategy introduces minimal cost. We evaluate the computational cost on the drjohnson scene to demonstrate our strategy's minimal overhead: We pre-compute the edge maps for all images at the start of training, which takes only 2.06s in total. During training, the average time spent on patch comparison is merely 0.0023s per iteration. To evaluate the impact on total training time, we compared our method with Taming-GS. With a Gaussian budget of 2.0M (ours) vs. 3.27M (Taming-GS), we measured the time per iteration once the Gaussian count stabilized: our method takes 0.0365s compared to 0.0460s for Taming-GS.
> These results demonstrate that our strategy introduces negligible overhead. More importantly, by maintaining a fewer but more efficient set of Gaussians, we actually reduce the overall training burden and memory footprint.
>
> Q4: Depth-gradient-based importance score (K4)
>
> We agree that depth continuity and consistency are critical indicators of reconstruction quality. Since depth-based gradients can reflect geometric misalignments that intensity edges might miss, we plan to incorporate a depth-gradient-based importance score in future work to more accurately guide the selection of candidate Gaussians for densification.
>
> Q5: Multi-view consistency (W2)
>
> We agree that artifact identification is performed per-view; however, since Gaussians are shared geometric primitives across all viewpoints, the optimization process inherently acts as a global consistency constraint. Throughout the training process, each Gaussian is progressively 'vetted' and calibrated by photometric constraints from disparate views. This ensures that any per-view modification is reconciled within the global scene representation, preventing localized adjustments from degrading overall multi-view integrity. Nonetheless, to further strengthen this consistency, we have discussed the potential for explicit multi-view constraints in our Conclusion (Lines 436-439). A feasible approach would be to record artifact pixel locations across views during early training; when a patch is identified, the dominant Gaussians could be back-projected into 3D space and then re-projected onto adjacent views to verify their spatial consistency. This would allow the system to confirm if a Gaussian consistently contributes to artifacts across multiple perspectives, thereby further refining the precision of our identification process.

---

> > ### Author Rebuttal · Reviewer_uGeV · 2026-04-04
> >
> > The authors have effectively addressed my primary concerns. The method is logically sound, the new timing details prove its efficiency, and the justifications for the heuristic choices are practical for a 3DGS pipeline. I will maintain my current score.

---

> > > ### Author Response · Authors · 2026-04-07
> > >
> > > Thanks for your acknowledge of our work. Thanks again for your time and effort in reviewing our paper.

---

### Official Review · Reviewer_oJiN · 2026-03-12

**Soundness:** 3
**Presentation:** 3
**Significance:** 3
**Originality:** 3
**Overall Recommendation:** 5
**Confidence:** 5

**Summary:**

This paper proposes PCGS, a Patch Comparison Gaussian Splatting method designed to mitigate blurring and artifacts in regions with overlapping Gaussians in 3D Gaussian Splatting. The approach observes that commonly used view-space positional gradients are insufficient for guiding densification in such challenging regions. To address this limitation, PCGS introduces a patch-based error analysis that identifies high-error image regions by comparing rendered outputs with ground truth images, and adaptively applies densification to the corresponding Gaussians. In addition, a Gaussian control strategy with a dynamically adjusted budget and importance-based sampling is proposed to prevent over-densification and redundant Gaussians. Experiments on several benchmarks show that the method reduces artifacts and blur while maintaining a Gaussian count comparable to that of standard 3DGS.

**Compliance With Llm Reviewing Policy:**

Affirmed.

**Final Justification:**

I am quite positive about the interesting observation and novel method presented in manuscript. Experiments in the manuscript and rebuttal show the method are simple but generally effective. And after reading other reviewer's comment, i still tend to keep clear accept for this submission. Therefore, i wanna champion for this work.

**Key Questions For Authors:**

Please refer to the weakness part.

**Limitations:**

limitation section is not included in this manuscript.

**Strengths And Weaknesses:**

### Strengths
1. The key observation of this work is quite interesting: there is a substantial imbalance in the contributions of individual Gaussians in scene reconstruction, which potentially leads to blurry regions in rendering.

2. The method design is direct and effective. The patch comparison is a sound design to mitigate patch-level blurring, alongside the Gaussian growth control mechanism to limit the total number of Gaussians.

3. The main experimental results presented in Tab. 2 and Fig. 4 strongly demonstrate the effectiveness of this work.

### Weaknesses
1. With the introduced Gaussian growth control mechanism, the number of Gaussians on different datasets should be reported in the experimental results, possibly included in the main results table (Tab. 2).

2. For reconstruction evaluation, it would be better to provide rendered videos to demonstrate the effectiveness of the proposed method.

---

> ### Author Rebuttal · Authors · 2026-03-28
>
> We appreciate the reviewer’s recognition of our key idea and experimental results. We sincerely thank the reviewer for the positive feedback and helpful suggestions.
>
> Q1: Number of Gaussians (W1)
>
> We agree that reporting the Gaussian count is important. In fact, we have already included comparisons of Gaussian numbers (Primes) and training time (Time) with other Gaussian-based methods in Table 3:
>
> | Dataset | Mip-NeRF360 |  | Tanks&Temples |  | Deep Blending |  |
> | :--- | :---: | :---: | :---: | :---: |:---: |:---: |
> | Method | Time | Primes | Time | Primes | Time | Primes |
> | 3DGS | 42m | 3178K | 27m | 1831K | 36m| 2805K|
> | Mini-GS | 26m | 485K | 16m | 303K | 22m| 557K|
> | Abs-GS | 33m | 3120K | 15m | 1304K | 37m| 3001K|
> | Pixel-GS | 47m | 5575K | 33m | 4490K | 47m| 4614K|
> | Perceptual-GS | 47m | 2746K | 26m | 1726K | 51m| 2887K|
> | Steep-GS | 28m | 2188K | 16m | 1309K | 28m| 1587K|
> | Taming-GS | 20m | 3207K | 12m | 1835K | 16m| 2800K|
> | Ours | 21m | 2500K | 13m | 1500K | 18m| 1750K|
>
> To improve clarity and make this information more accessible, we will move these statistics to the main results table (Tab. 2) in the final version.
>
> Q2: Video results (W2)
>
> We agree that rendered videos would provide more comprehensive evaluation. We will include video results in the supplementary material in the final version.
>
> Q3: Limitations
>
> We thank the reviewer for this constructive suggestion. As noted in the conclusion (Lines 436-439), our current pipeline primarily leverages view-dependent optimization, which may occasionally struggle with complex view-independent artifacts. To address this, we agree that incorporating multi-view consistency and geometric priors is a promising direction.
>
> Specifically, in future work, we plan to integrate auxiliary signals such as depth-based geometry constraints, perceptual priors, and semantic consistency to achieve more robust localization and elimination of floaters. To ensure these points are clear to the readers, we will move this discussion into a dedicated 'Limitations and Future Work' section in the final version and provide a more in-depth analysis.

---

> > ### Author Rebuttal · Reviewer_oJiN · 2026-04-03
> >
> > I appreciate authors detailed response, i will maintain the initial rating.

---

> > > ### Author Response · Authors · 2026-04-07
> > >
> > > Thanks for your acknowledge of our work. Thanks again for your time and effort in reviewing our paper.

---

### Decision · Program_Chairs · 2026-04-30

**Decision:**

Accept (regular)

**Comment:**

The paper received unanimously positive ratings, with 1 Accept and 3 Weak Accepts. The reviewers recognized the paper's well-motivated analysis of blurring artifacts in 3DGS. While initial concerns were raised regarding the reliance on specific heuristics, the absence of explicit multi-view consistency constraints, and the limited novelty of the growth control module, the authors successfully addressed these critiques during the rebuttal phase.

Following the rebuttal, all reviewers reached a positive consensus, agreeing that the proposed method is technically sound and delivers notable rendering improvements. Having reviewed the paper, the rebuttal, and the reviewer feedback, the AC aligns with this consensus and recommends acceptance.